# Resource-Efficient Reinforcement for Reasoning Large Language Models via Dynamic One-Shot Policy Refinement

**Yunjian Zhang** [* 1]  **Sudong Wang** [* 2]  **Yang Li** [3]  **Peiran Xu** [4]  **Conghao Zhou** [5]  **Xiaoyue Ma** [6]  **Jianing Li** [7]  **Yao Zhu** [8]

## Abstract

Large language models (LLMs) have exhibited remarkable performance on complex reasoning tasks, with reinforcement learning under verifiable rewards (RLVR) emerging as a principled framework for aligning model behavior with reasoning chains. Despite its promise, RLVR remains prohibitively resource-intensive, requiring extensive reward signals and incurring substantial rollout costs during training. In this work, we revisit the fundamental question of data and compute efficiency in RLVR. We first establish a theoretical lower bound on the sample complexity required to unlock reasoning capabilities, and empirically validate that strong performance can be achieved with a surprisingly small number of training instances. To tackle the computational burden, we propose Dynamic One-Shot Policy Refinement (DoPR), an uncertainty-aware RL strategy that dynamically selects a single informative training sample per batch for policy updates, guided by reward volatility and exploration-driven acquisition. DoPR reduces rollout overhead by nearly an order of magnitude while preserving competitive reasoning accuracy, offering a scalable and resource-efficient solution for LLM post-training. This approach offers a practical path toward more efficient and accessible RL-based training for reasoning-intensive LLM applications.

---

[*]Equal contribution [1]University of Chinese Academic of Sciences [2]The Hong Kong University of Science and Technology (GZ) [3]Tsinghua University [4]Sun Yat-Sen University [5]Xidian University [6]George Mason University [7]Peking University [8]Zhejiang University. Correspondence to: Yao Zhu <ee_zhuy@zju.edu.cn>, Xiaoyue Ma <xma9@gmu.edu>.

*Proceedings of the 43rd International Conference on Machine Learning*, Seoul, South Korea. PMLR 306, 2026. Copyright 2026 by the author(s).

## 1. Introduction

Recent advances in large language models (LLMs) (Shao et al., 2024; DeepSeek-AI et al., 2025; Liu et al., 2025; Achiam et al., 2023; Hu et al., 2025; Ma et al., 2025) have led to significant progress in solving complex reasoning tasks. A key driver behind these advances is the adoption of reinforcement learning (RL) (Zhang et al., 2025a; Lin et al., 2025; Xiong et al., 2025), which has proven effective in activating coherent reasoning trajectories within LLMs. Leading LLMs, including GPT-o1 (Contributors et al., 2024), Gemini (Reid et al., 2024), and DeepSeek-r1 (DeepSeek-AI et al., 2025), all incorporate RL-based post-training stages to refine the models' reasoning abilities beyond what standard supervised fine-tuning (SFT) (Chen et al., 2024; Tajwar et al., 2024; Dong et al., 2023) can achieve. These developments underscore the growing importance of RL as a tool for enhancing the reasoning performance of LLMs.

Reinforcement learning with verifiable rewards (RLVR) (Gao et al., 2024; Lambert et al., 2024; Team et al., 2025; Wang et al., 2025b;a) has emerged as a prominent paradigm for training reasoning language models, which provides binary and verifiable feedback indicating the correctness of the model's answer. Recent work has primarily focused on improving the performance of RL algorithms (Chen et al., 2025b; Yue et al., 2025; Xu et al., 2025; Cui et al., 2025), while comparatively little attention has been paid to their resource demands. From a data perspective, RLVR requires large quantities of samples with high-quality reward signals to drive effective learning. On the computational side, RLVR typically performs multiple rollouts over the entire dataset during training to ensure stable and effective gradient-based policy optimization, leading to substantial computational overhead. Although some work aims at improving training efficiency (Wang et al., 2025b; Zhao et al., 2025), they rely on extensive pre-training over large datasets to identify the most suitable training instances, thereby failing to fundamentally reduce resource consumption.

This work aims to reduce the data and computational overhead of RLVR during the post-training of reasoning LLMs. We begin by deriving a theoretical lower bound on the sample complexity required to elicit optimal reasoning behavior

during RLVR. Surprisingly, our empirical findings reveal that even with a remarkably small number of training instances, the model can achieve near-optimal reasoning performance. This suggests that RL supervision serves more as a capability activator rather than a performance booster, and that the reasoning ability of the model is largely constrained by its pre-training rather than the volume of RL data. These results challenge the prevailing assumption that large-scale rewards are necessary, and instead highlight the potential of minimalist, data-efficient reinforcement strategies. These insights enable us to drastically reduce the training dataset to a compact subset without compromising final performance. In parallel, we introduce Dynamic One-Shot Policy Refinement (DoPR), a novel reinforcement learning strategy that dynamically selects a single most informative instance within each mini-batch. By focusing policy updates on maximally impactful samples, DoPR significantly reduces the number of rollouts, yielding substantial savings in computational cost while maintaining comparable performance.

Our contributions can be summarized as follows:

- We present the first systematic investigation into the nexus between data scale and reasoning performance within LLMs. By establishing a rigorous theoretical lower bound on sample complexity, we demonstrate that reasoning capabilities under RLVR can be activated by a minimalist data regime. This finding fundamentally decouples optimal reasoning performance from large-scale data requirements, challenging the long-standing paradigm that massive supervision is indispensable for effective RL alignment.

- We propose DoPR, a lightweight yet effective strategy that leverages historical rewards to select a single high-value training instance per batch for policy updates. This approach substantially reduces rollout overhead by nearly an order of magnitude while preserving reasoning performance, enabling efficient and cost-effective RL training for reasoning tasks.

- Extensive experiments reveal that our approach is capable of reducing the training dataset to as few as 16 instances and constraining the rollout budget to a single sample per batch, while maintaining competitive reasoning performance.

## 2. Related Work

### 2.1. Reasoning Large Language Models

Human-like reasoning has received growing attention for its potential to support generalization across abstract, multi-step tasks (Kojima et al., 2022; Zhou et al., 2023). Early efforts primarily focus on prompting methods to elicit latent reasoning capabilities from pre-trained LLMs. For example,

Chain-of-Thought (CoT) prompting (Wei et al., 2022) enables step-by-step reasoning without any additional training, while more structured paradigms such as Tree-of-Thought (ToT) (Yao et al., 2023) and Graph-of-Thought (GoT) (Besta et al., 2023) incorporate hierarchical or graph-based reasoning paths. In addition, self-consistency decoding (Wang et al., 2023) enhances reliability by aggregating multiple sampled reasoning traces.Some efforts have investigated more effective training strategies to improve the model's intrinsic reasoning competence. For instance, LIMO (Ye et al., 2025) applies SFT on curated mathematical reasoning datasets to explicitly guide the model toward correct solution steps. Moreover, some researchers utilize LLM-driven search algorithms to automatically generate accurate reasoning trajectories through trial-and-error search (Zheng et al., 2023), and then train Process Reward Models (PRMs) on these reasoning trajectories (Uesato et al., 2022), which provide dense and intermediate rewards to enable reinforcement learning over reasoning chains.

### 2.2. Reinforcement Learning with Verifiable Rewards

RLVR has emerged as a principled framework for enhancing reasoning capabilities, particularly in domains where correctness can be objectively assessed, such as mathematical problem solving. Instead of relying on human feedback or learned reward models, RLVR employs rule-based verification mechanisms to generate reward signals, enabling robust optimization of reasoning policies with minimal annotation cost. The effectiveness of RLVR is first demonstrated by OpenAI's o1 series (Contributors et al., 2024), and subsequent models such as DeepSeek-R1 (DeepSeek-AI et al., 2025) and the Qwen series (Bai et al., 2023) further advance the use of verifiable feedback in training reasoning LLMs. On the algorithmic front, GRPO (Shao et al., 2024; DeepSeek-AI et al., 2025) and its extensions (Liu et al., 2025; Zhang et al., 2025b; Kong et al., 2025) have become foundational in enabling efficient reward-driven training. Follow-up frameworks such as DAPO (Yu et al., 2025), VAPO (Yue et al., 2025), SimpleRLZoo (Zeng et al., 2025), and Open-Reasoner-Zero (Hu et al., 2025) further explore various design choices for policy optimization, reward shaping, and data reuse under verifiable reward settings.

Despite the notable successes, existing RLVR strategies suffer from substantial resource demands. First, they typically require a large amount of reward signals to supervise policy learning. Although some efforts (Zhao et al., 2025; Wang et al., 2025b) reduce the number of training samples, they still rely on preselection throughout the dataset and thus do not fundamentally reduce the effective sample size. Second, GRPO-based algorithms incur significant computational overhead during training, as they rely on repeated rollouts over the entire dataset to estimate gradients and improve policies. This work takes resource efficiency as the central

goal, aiming to reduce the data and compute demands of RLVR without sacrificing its reasoning performance.

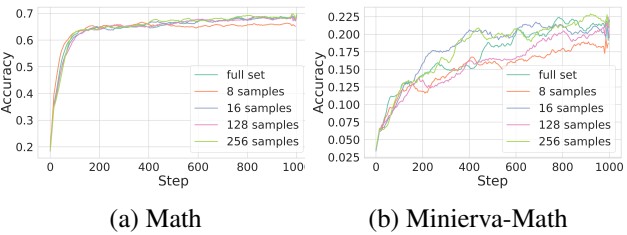

(a) Math          (b) Minierva-Math

*Figure 1.* Accuracy on MATH and Minierva-MATH with varying training set sizes. Except for the 8-sample setting, all configurations converge to comparable performance, indicating that strong reasoning ability can be achieved with few training examples.

## 3. Rethinking Data Requirements for RLVR

We begin by revisiting a fundamental question: *How much data is truly required to unlock the reasoning capabilities of LLMs through RLVR?* Prior studies have explored various strategies for reducing the training data scale in RLVR (Wang et al., 2025b; Ye et al., 2025). These methods typically rely on carefully crafted generation or selection strategies to identify a subset of high-quality samples for reinforcement learning. While such heuristics can be empirically effective, they provide only a pragmatic workaround rather than a principled understanding of the relationship between data volume and policy quality. Crucially, they do not address the core question of *whether strong reasoning performance can be achieved with limited reward signals.* Beyond heuristic-based sample selection, we investigate whether a small number of training instances can provably yield non-trivial gains in reasoning performance under verifiable rewards. We ground our analysis in the policy gradient formulation used in RLVR, and derive a lower bound of required data scale on expected policy improvement.

For a pre-training policy $\pi_{\theta_0}$, its performance gap with the optimal policy $\pi_{\theta^\star}$ on a reasoning task is:

$$\mathbb{E}_{x\sim\mathcal{D}}[P^{\pi_{\theta^\star}}(x) - P^{\pi_{\theta_0}}(x)] = \epsilon, \quad (1)$$

where $x$ is an instance sampled from the dataset $\mathcal{D}$, and $P^\pi(x)$ denotes the expected verification reward.

**Theorem 3.1.** *Consider an optimization procedure where the policy is updated using a single sample at each step, let $\pi_{\theta^\star}$ denotes the optimal policy and $\pi_{\theta_N}$ denotes the policy after $N$ updates. To guarantee that the expected performance gap satisfies*

$$\mathbb{E}_{x\sim\mathcal{D}}\left[P^{\pi_{\theta^\star}}(x) - P^{\pi_{\theta_N}}(x)\right] \leq \epsilon', \quad (2)$$

*it suffices that the number of steps $N$ satisfies*

$$N \geq \mathcal{O}(\ln\frac{\epsilon}{\epsilon'}), \quad (3)$$

*where $\epsilon$ denotes the initial performance gap.*

*Proof.* Let $J(\theta) = \mathbb{E}_{x\sim\mathcal{D}}[P^{\pi_\theta}(x)]$ denote the expected performance of the policy parameterized by $\theta$. Assume that $J$ is locally $L$-smooth (Chen et al., 2025a), i.e., there exists a constant $L > 0$ and a neighborhood $\mathcal{N}$ containing the initial policy $\theta_0$, such that the following Lipschitz inequality holds for all $\theta, \theta' \in \mathcal{N}$:

$$J(\theta') \geq J(\theta) + \langle\nabla J(\theta), \theta' - \theta\rangle - \frac{L}{2}\|\theta' - \theta\|^2. \quad (4)$$

We remark that this assumption holds primarily in the vicinity of the optimization trajectory, and its validity is empirically ensured by conservative step sizes, KL regularization, and gradient clipping. By the policy gradient theorem, $J$'s gradient is given by:

$$\nabla J(\theta) = \mathbb{E}_{x\sim\mathcal{D}}\left[\nabla_\theta P^{\pi_\theta}(x)\right]. \quad (5)$$

Assume that the policy is updated at each step via gradient ascent with a fixed learning rate $\alpha > 0$:

$$\theta_{t+1} = \theta_t + \alpha g_t, \quad (6)$$

where $g_t$ is an unbiased stochastic estimate of the true gradient, i.e., $\mathbb{E}[g_t] = \nabla J(\theta_t)$, and the variance of $g_t$ is bounded by $Var(g_t) \leq \delta^2$. Taking expectations and applying the smoothness of $J$, we obtain:

$$\mathbb{E}\left[J(\theta_{t+1}|\theta_t)\right] \geq J(\theta_t) + \alpha(1 - \frac{L\alpha}{2})\|\nabla J(\theta_t)\|^2 - \frac{L\alpha^2\delta^2}{2}, \quad (7)$$

Pretraining and supervised fine-tuning (SFT) endow the model with an initial reasoning capability. Coupled with the explicit gradient signal provided by RLVR, the objective $J$ satisfies a local Polyak–Łojasiewicz (PL) condition in the neighborhood induced by pretraining and SFT (Aich et al., 2025; Peng et al., 2024). Intuitively, the combination of pretraining/SFT (which shapes a favorable local manifold) and the supervised RLVR gradients renders the landscape locally well-conditioned for efficient optimization. That is, $J$ satisfies

$$\|\nabla J(\theta_t)\|^2 \geq c\left(J(\theta^\star) - J(\theta_t)\right) = c\Delta_t, \quad (8)$$

for some constant $c > 0$, where $\theta^\star$ is the optimal parameter and $\Delta_t = J(\theta^\star) - J(\theta_t)$ denotes the performance gap at step $t$. Then, we have:

$$\mathbb{E}\left[J(\theta_{t+1}) - J(\theta_t) \mid \theta_t\right] \geq \alpha c(1 - \frac{L\alpha}{2})\Delta_t - \frac{L\alpha^2\delta^2}{2}. \quad (9)$$

Let $\eta = \alpha c/2$, and choose $\alpha$ such that $1 - \frac{L\alpha}{2} \geq \frac{1}{2}$, we further require

$$\frac{L\alpha^2\delta^2}{2} \leq \frac{\eta\epsilon'}{2},$$

which leads to the recursive inequality:

$$\mathbb{E}\left[\Delta_{t+1} \mid \theta_t\right] \leq (1 - \eta)\Delta_t + \frac{\epsilon'}{2}. \qquad (10)$$

Unfolding this recurrence, we obtain:

$$\mathbb{E}\left[\Delta_t\right] \leq (1 - \eta)^t \Delta_0 + \frac{\epsilon'}{2}, \qquad (11)$$

where $\Delta_0 = J(\theta^\star) - J(\theta_0) = \epsilon$ is the initial gap. To ensure $\mathbb{E}\left[\Delta_t\right] \leq \epsilon'$, it suffices that:

$$(1 - \eta)^t \epsilon \leq \frac{\epsilon'}{2}. \qquad (12)$$

Taking logarithms on both sides yields:

$$t \geq \frac{\ln(\epsilon'/2) - \ln(\epsilon)}{\ln(1 - \eta)} = \mathcal{O}(\ln \frac{\epsilon}{\epsilon'}), \qquad (13)$$

For more detailed derivation, please refer to the appendix. $\qquad \square$

In practical applications, the initial policy (such as the Qwen-math series) is typically a version that has already undergone supervised fine-tuning, which inherently possesses a certain degree of reasoning capability. Consequently, the initial performance gap is typically small, implying that $\ln(\frac{\epsilon}{\epsilon'})$ remains modest. Therefore, convergence can be achieved within a constant number of training steps.

To assess the impact of training data size on RLVR performance, we conduct a series of experiments using the Qwen2.5-Math-1.5B model (Yang et al., 2024) on the MATH (Hendrycks et al., 2021) and Minerva-MATH (Lewkowycz et al., 2022) benchmarks, varying the number of training samples across five settings: 8, 16, 128, 256, and the full dataset (1209 examples). As shown in Figure 1, we observe that all configurations with 16 or more samples converge to nearly identical validation accuracy. This result corroborates our theoretical analyses, and provides empirical evidence that only a small subset of the data is sufficient to elicit strong reasoning capabilities in RLVR training. Our analyses reveal that substantial reductions in training data are possible without sacrificing final performance, paving the way for more efficient RL-based reasoning.

## 4. Dynamic One-Shot Policy Refinement

Currently, reasoning model training often builds upon GRPO and its variants, which leverage group-wise relative rewards to estimate the advantages over the baseline. In particular, GRPO circumvents value function approximation by computing the average reward over multiple responses to the same input query, using this group-level statistic as a baseline for advantage estimation. For each question $q$, GRPO samples a group of $G$ responses $\{o_i\}_{i=1}^G$ from the

---

**Algorithm 1** Dynamic One-Shot Policy Refinement (DoPR)

**Require:** Initial policy $\pi_{\theta_0}$, reference policy $\pi_{\text{ref}}$, rollout budget $G$, batch size $K$, momentum factors $\rho_1$, $\rho_2$, exploration weight $\lambda$, total steps $T$

1: Initialize reward statistics $\mu_i^0 \leftarrow 0$, $\sigma_i^{0,2} \leftarrow 0$, and selection counter $n_i \leftarrow 0$ for each sample $x_i$ in the dataset $\mathcal{D}$
2: **for** $t = 1$ to $T$ **do**
3:     Sample a mini-batch $\mathcal{B} = \{o_1, \ldots, o_K\}$ from the dataset
4:     **for all** $x_i \in \mathcal{B}$ **do**
5:         Perform one rollout $o_i^t \sim \pi_{\theta_t}(x_i)$
6:         Compute scalar reward $r_i^t$
7:         Update exponential moving average of mean and variance:
8:         $\mu_i^t \leftarrow \rho_1 \cdot r_i^t + (1 - \rho_1) \cdot \mu_i^{t-1}$
9:         $\sigma_i^{t,2} \leftarrow \rho_2 \cdot (r_i^t - \mu_i^t)^2 + (1 - \rho_2) \cdot \sigma_i^{t-1,2}$
10:       Compute EM-UCB exploration term: $U_i^t \leftarrow$ Sigmoid$\left(\frac{H_i^t - \mu_H}{\delta_H + \epsilon}\right) \cdot \sqrt{\frac{\log(t+1)}{n_i + 1}}$
11:         Compute acquisition score: $S_i^t \leftarrow \sigma_i^t + U_i^t$
12:     **end for**
13:     Select top-scoring instance: $o^* \leftarrow \arg\max_{o_i \in \mathcal{B}} S_i^t$
14:     Increment selection count: $n_{o^*} \leftarrow n_{o^*} + 1$
15:     Perform $G$ rollouts on $o^*$: $\{o_j\}_{j=1}^G \sim \pi_{\theta_t}(o^*)$
16:     Compute group rewards and token-wise advantages $\hat{A}_{j,t}$ for GRPO
17:     Compute policy gradient loss $\mathcal{L}_{\text{GRPO}}(\theta)$
18:     Update policy $\theta_{t+1} \leftarrow \theta_t - \eta \nabla_\theta \mathcal{L}_{\text{GRPO}}(\theta)$
19: **end for**
**output** Trained policy $\pi_{\theta_T}$

---

current policy $\pi_{\theta_{old}}$, and optimizes the policy by maximizing the expected advantage over these responses:

$$\mathcal{J}_{\text{GRPO}}(\theta) = \mathbb{E}_{q \sim P(Q), \{o_i\}_{i=1}^G \sim \pi_{\theta_{\text{old}}}} \left[ \frac{1}{G} \sum_{i=1}^G \frac{1}{|o_i|} \sum_{t=1}^{|o_i|} \Big\{ \right.$$

$$\left. \min\left[ \mathcal{F}(\pi) \hat{A}_{i,t}, \text{ clip}_{1-\epsilon}^{1+\epsilon}(\mathcal{F}(\pi)) \hat{A}_{i,t} \right] - \beta \, \mathbb{D}_{\text{KL}}(\pi_\theta \parallel \pi_{\text{ref}}) \Big\} \right], \qquad (14)$$

$$\mathcal{F}(\pi) = \frac{\pi_\theta(o_{i,t}|q, o_{i,<t})}{\pi_{\theta_{old}}(o_{i,t}|q, o_{i,<t})}, \qquad (15)$$

where $\hat{A}_{i,t}$ is computed based on the group reward scores of the $t$-th token.

For each query in the batch, GRPO performs $G$ rollouts to estimate group-wise rewards, which are subsequently used to compute response-level advantages. Given a batch size of $K$, each policy update necessitates $K \times G$ rollouts, resulting in substantial computational overhead. To address this issue, we introduce Dynamic One-Shot Policy Refinement (DoPR), a lightweight and computationally efficient strat-

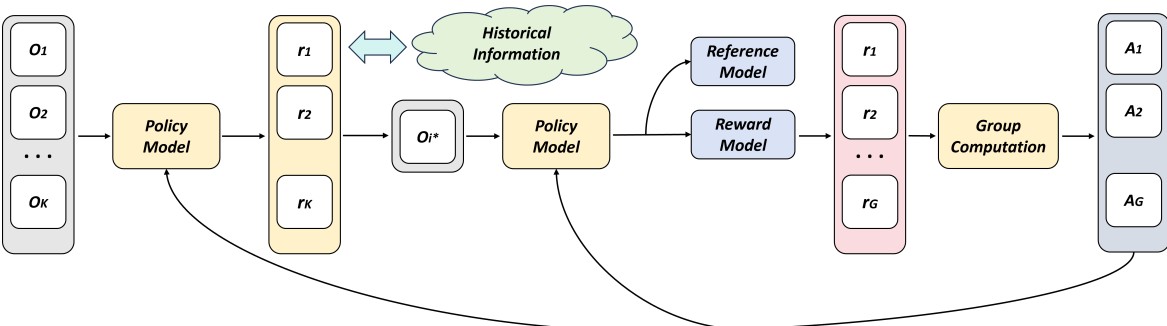

*Figure 2.* Overview of DoPR, which dynamically selects a single high-value training instance from each mini-batch based on historical reward statistics.

egy that selectively refines the policy using only a single high-influence training instance, as shown in Figure 2.

For each sample $o_i$ in a mini-batch $\{o_i\}_{i=1}^K$, we maintain a historical record of the reward statistics obtained from previous rollouts, including the mean $\mu_i$ and variance $\sigma_i^2$. In the $t$-th step, we execute a single rollout for $o_i$ using the current policy $\pi_{\theta_t}$, yielding a reward $r_i^t$, then calculate the exponentially weighted momentum estimates of the reward:

$$\begin{aligned} \mu_i^t &= \rho_1 \cdot r_i^t + (1 - \rho_1) \cdot \mu_i^{t-1}, \\ \sigma_i^{t,2} &= \rho_2 \cdot (r_i^t - \mu_i^t)^2 + (1 - \rho_2) \cdot \sigma_i^{t-1,2}, \end{aligned} \quad (16)$$

where $\rho_1$ and $\rho_2$ are momentum hyperparameters controlling the temporal sensitivity. These momentum statistics allow us to capture the reward volatility of each sample over time, providing a principled measure of uncertainty under the current policy.

To address the exploration–exploitation trade-off, we propose an Entropy-Modulated Upper Confidence Bound (EM-UCB) acquisition score. Standard UCB (Kaelbling, 1994; Garivier & Moulines, 2011) treats all samples uniformly, inducing unnecessary exploration even when the policy has already converged on stable predictions. EM-UCB remedy this limitation by incorporating the policy entropy as a confidence-aware gate. As a result, exploratory updates are emphasized only for samples where the policy remains uncertain, aligning exploration with potential information gain, leading to more efficient rollout utilization without introducing additional training complexity.

$$U_i^t = \text{Sigmoid}\left(\frac{H_i^t - \mu_H}{\delta_H + \epsilon}\right) \cdot \sqrt{\frac{\log(t+1)}{n_i + 1}}, \quad (17)$$

$$H_i^t = -\frac{1}{\mathcal{T}} \sum_{l=1}^{\mathcal{T}} \sum_{v \in \mathcal{V}} \pi_\theta(v|o_i, < l) \log \pi_\theta(v|o_i, < l), \quad (18)$$

where $n_i$ denotes the cumulative number of times sample $o_i$ has been selected, $\mu_H$, $\delta_H$ are the running mean and standard deviation of entropy values across the batch, $\mathcal{T}$

denotes the output length, and $\mathcal{V}$ is the vocabulary. The resulting EM-UCB term adaptively regulates exploration: when the policy exhibits high uncertainty (large $H_i^t$), the exploration bonus is amplified; when the policy is confident, the UCB contribution is suppressed, preventing unnecessary exploration on already-determined samples. Therefore, the composite acquisition score for each sample is defined as:

$$S_i^t = \sigma_i^t + U_i^t. \quad (19)$$

$$i^t = \arg\max_i S_i^t. \quad (20)$$

This scoring function balances exploitation (selecting samples with high reward variance) and exploration (favoring underused high-entropy samples). The core behind the exploitation is that the model is more uncertainty about the samples with high reward variance and high entropy, which are likely to be more informative for policy refinement.

After selecting the most informative sample $o_{i^t}$, we allocate the full rollout budget $G$ on it, collecting $G$ responses $\{o_{i^t,j}\}_{j=1}^G$ from the current policy $\pi_{\theta_t}$. The policy is then updated using the group-wise relative rewards of these responses, following the GRPO formulation in Eq.(14).

In contrast to the conventional GRPO that requires $K \times G$ rollouts per step, DoPR reduces the rollout cost to $G + (K-1)$, where $G$ rollouts are performed only on the selected sample and $(K-1)$ rollouts are performed on the remaining. This results in nearly an order-of-magnitude reduction in rollout cost, while still maintaining effective policy learning through targeted updates. DoPR's design is simple, general, and easily integrable into existing RLVR pipelines. The pseudo code of DoPR is presented in Algorithm 1.

## 5. Experiments

### 5.1. Experimental Settings

We conduct experiments using the Qwen2.5-Math and LLaMA3.1 (Yang et al., 2024) as the base language models, which are widely used for mathematical reasoning tasks. For

*Table 1.* Pass@1 performance comparison across multiple mathematical reasoning benchmarks.

| Method (Total Rollouts Fixed) | AIME24 | AMC | MATH | MIN. | OLY. | GSM8K | Avg. |
|---|---|---|---|---|---|---|---|
| *Baseline methods* | | | | | | | |
| Qwen2.5-base **1.5B** | 0.0 | 0.0 | 3.2 | 2.2 | 2.4 | 3.9 | 1.9 |
| Qwen2.5-base **7B** | 13.3 | 45 | 53.4 | 15.1 | 27.4 | 59.1 | 35.6 |
| Qwen2.5-Math-base **1.5B** | 3.3 | 25.0 | 22.6 | 6.2 | 13.3 | 23.5 | 15.0 |
| Qwen2.5-Math-base **7B** | 10 | 42.5 | 44.6 | 12.1 | 14.4 | 53.4 | 29.5 |
| GRPO Qwen **1.5B** | 20.0 | 60.0 | 71.8 | 29.8 | 34.2 | 82.0 | 49.6 |
| GRPO Qwen **7B** | 26.7 | 65 | 78.2 | 29.4 | 37.9 | 87.6 | 54.1 |
| GRPO LLaMA **8B** | 3.3 | 10.0 | 25.8 | 11.4 | 6.5 | 70.9 | 21.3 |
| One-Shot RL Qwen **1.5B** | 13.3 | 55.2 | 66.6 | 19.5 | 29.8 | 76.6 | 43.5 |
| One-Shot RL Qwen **7B** | 16.7 | 52.5 | 74.0 | 23.9 | 32.1 | 81.8 | 46.8 |
| One-Shot RL LLaMA **8B** | 0.0 | 15.0 | 28.0 | 16.9 | 7.7 | 72.8 | 17.6 |
| UFO Qwen **1.5B** | 19.4 | 59.2 | 73.3 | 29.7 | 33.5 | 81.7 | 49.4 |
| UFO Qwen **7B** | 27.9 | 52.8 | 73.5 | 38.7 | 38.2 | 85.3 | 52.7 |
| UFO LLaMA **8B** | 3.1 | 14.4 | 26.8 | 16.4 | 8.1 | 72.6 | 23.6 |
| *Our method* | | | | | | | |
| DoPR Qwen **1.5B** | 19.7 | 59.3 | 73.5 | 29.4 | 33.9 | 82.2 | 49.6 |
| DoPR Qwen **7B** | 30.0 | 52.5 | 73.8 | 39.0 | 37.9 | 85.6 | 53.1 |
| DoPR LLaMA **8B** | 3.3 | 15.0 | 28.0 | 16.9 | 7.7 | 72.8 | 24.0 |

training, we adopt the same setting with (Wang et al., 2025b), which randomly selects a subset of 1209 high-quality reasoning examples from the DeepScaleR-Preview-Dataset (Luo et al., 2025) to construct the training set, and use GRPO as the policy optimization backbone under consistent rollout configurations. The evaluation is performed on seven widely used mathematical reasoning benchmarks, including AIME24, AMC, MATH (Hendrycks et al., 2021), Minerva Math (Lewkowycz et al., 2022), OlympiadBench (Huang et al., 2024), and GSM8K (Cobbe et al., 2021). The evaluation metric is Pass@1 accuracy. Since AIME24 contains only 30 problems and a single run yields highly volatile estimates, we report avg@16 on AIME24, i.e., the mean Pass@1 over 16 independent samples per question with temperature $0.6$ and top-$p$ $0.95$, to obtain a more reliable estimate. We choose UFO RL (Zhao et al., 2025), One-Shot RL (Wang et al., 2025b) without entropy strategy, and standard GRPO (Shao et al., 2024) as the baselines, and the number of total training steps for all methods is set to 1000. From Figure 1, we observe that the sample size of 16 is sufficient to attain competitive performance, thus we adopt the 16-sample configuration for DoPR in experiments.

### 5.2. Benchmarking Across Diverse Tasks

Table 1 presents the pass@1 performance of various reinforcement learning methods. We observe that RLVR brings substantial improvements over the base model. For instance on the Qwen2.5 1.5B model, GRPO outperforms the base model by a large margin across all datasets, improving the average accuracy from 37.8 to 49.6. This demonstrates

the effectiveness of outcome-based policy optimization in enhancing reasoning capabilities. Our proposed method, DoPR, achieves comparable performance to GRPO with an average accuracy of 49.6, while requiring significantly fewer samples and rollouts during training, confirming its efficiency without sacrificing effectiveness. We also observe that One-Shot RL underperforms both GRPO and DoPR across most benchmarks. This performance gap can be attributed to its reliance on single-response updates, which provide limited gradient information and may lead to unstable training dynamics. In our experiments, One-Shot RL often exhibits early saturation and struggles to make further progress in later training stages due to gradient degradation. In contrast to uniform or random sampling approaches, DoPR dynamically identifies and exploits informative training instances to refine the policy, resulting in consistently stable learning behavior and improved sample efficiency throughout training.

### 5.3. Impact of Training Data Scale

We compare the performance of GRPO and DoPR with different scale of training data on Qwen 1.5B, and the results are shown in Table 2. It can be observed that both GRPO and DoPR demonstrate remarkable robustness to data reduction. Specifically, performance remains largely stable when using 16 or more samples, with minimal degradation compared to training on the full dataset. For example, GRPO trained with only 16 samples achieves 69.8 on MATH and 81.5 on GSM8K, closely matching the full-data performance of 71.8 and 82.0, respectively. A comparable pattern is

*Table 2.* Pass@1 accuracy under varying sample budgets in RLVR training.

| Method | AIME24 | AMC | MATH | MIN. | OLY. | GSM8K | Avg. |
|---|---|---|---|---|---|---|---|
| *Baseline methods* | | | | | | | |
| GRPO (Full data) | 20.0 | 60.0 | 71.8 | 29.8 | 34.2 | 82.0 | 49.6 |
| GRPO (256 samples) | 19.7 | 57.7 | 70.8 | 28.7 | 34.1 | 82.6 | 48.9 |
| GRPO (128 samples) | 20.0 | 57.5 | 71.7 | 29.8 | 32.1 | 82.7 | 48.9 |
| GRPO (16 samples) | 19.3 | 58.5 | 69.8 | 28.2 | 32.9 | 81.5 | 48.4 |
| GRPO (8 samples) | 10.0 | 50.0 | 69.8 | 25.9 | 31.3 | 77.0 | 44.0 |
| *Our methods* | | | | | | | |
| DoPR (128 samples) | 19.9 | 60.1 | 73.0 | 27.8 | 32.7 | 82.1 | 49.3 |
| DoPR (16 samples) | 19.7 | 59.3 | 73.5 | 29.4 | 33.9 | 82.2 | 49.6 |
| DoPR (8 samples) | 9.3 | 44.7 | 68.9 | 26.4 | 30.8 | 77.4 | 42.9 |

observed with DoPR, which also delivers strong results under the same data constraints. This suggests that effective policy learning via RLVR does not necessarily require extensive training corpora. However, when the training set is reduced to only 8 samples, a consistent drop in performance is observed across all benchmarks. This suggests a critical threshold below which the diversity and quantity of training data are insufficient to elicit the model's full reasoning potential. Importantly, the performance trend remains consistent across both optimization strategies, implying that the underlying data requirement characteristics are shared across GRPO and DoPR. These results validate our theoretical insights regarding the minimal data requirement for RLVR, and further demonstrate that DoPR can retain high performance even in low-resource regimes, offering a practical solution for data-efficient reasoning model training.

### 5.4. Comparison under Equal Rollout Budgets

To evaluate the rollout efficiency of different reinforcement learning strategies, we conduct experiments under fixed rollout budgets ranging from 5k to 50k. Table 3 summarizes the Pass@1 accuracy across various reasoning benchmarks for GRPO, One-Shot RL, and DoPR. It can be seen that under constrained rollout budgets, DoPR demonstrates a substantial performance advantage over GRPO. For instance, with only 10k rollouts, DoPR achieves comparable Pass@1 accuracy relative to GRPO trained with 50k rollouts. This underscores the inefficiency of GRPO's rollout strategy, which fails to distinguish high-impact samples from less informative ones during training. Consequently, a larger number of rollouts is required to reach similar levels of reasoning capability. In contrast, DoPR leverages a dynamic selection mechanism to identify and focus on the most informative samples at each iteration, guided by a composite score that captures both reward uncertainty and exploration value. This utility-aware update strategy ensures that each rollout contributes more effectively to policy improvement, enabling faster convergence and stronger performance with

significantly reduced computational cost. One-Shot RL also achieves competitive performance in extremely low-budget settings (e.g., 5k–10k rollouts). However, it fails to improve further as the rollout budget increases, due to its reliance on single-instance updates and lack of adaptive sample prioritization. In contrast, DoPR continues to scale effectively with larger budgets, eventually reaching performance parity with GRPO while maintaining superior efficiency throughout the training process. This reveals the core strength of DoPR: its ability to maximize the utility of limited rollout resources through principled and information-aware update scheduling. This makes it a practical and scalable solution for reinforcement learning in settings where rollout and computational efficiency are critical.

### 5.5. Ablation Studies

We conduct an ablation study to investigate the effectiveness of the EM-UCB term. As shown in Table 4, all methods exhibit similar performance after converging, suggesting that the ultimate reasoning capability is primarily determined by the underlying reinforcement learning framework itself, and that the choice of individual training samples per update has limited influence on the final performance ceiling. In contrast, when the rollout budget is constrained, full DoPR substantially outperforms both ablated variants. This demonstrates the advantage of the proposed EM-UCB strategy in improving learning efficiency by directing rollouts toward more informative instances.

### 5.6. Efficiency in Rollout and Update Time

We further evaluate the computational efficiency of DoPR by analyzing two key metrics: the average response length and the per-step update time in training. As shown in Figure 3, GRPO maintains a relatively constant response length across training steps, reflecting its fixed group sampling strategy that does not adapt to the model's evolving confidence. In contrast, DoPR exhibits a rapid decline in response length over the course of training, eventually stabilizing at

*Table 3.* Pass@1 accuracy under equal total rollout budget across different reinforcement learning strategies.

| Method (Total Rollouts Fixed) | AIME24 | AMC | MATH | MIN. | OLY. | GSM8K | Avg. |
|---|---|---|---|---|---|---|---|
| *Rollout budget: 5k* | | | | | | | |
| GRPO | 6.7 | 37.5 | 40.6 | 10.3 | 23.7 | 47.8 | 27.8 |
| One-Shot RL | 8.5 | 41.2 | 62.9 | 15.3 | 26.6 | 70.5 | 37.5 |
| DoPR | 14.6 | 46.0 | 66.4 | 20.9 | 30.1 | 75.2 | 42.2 |
| *Rollout budget: 10k* | | | | | | | |
| GRPO | 10.0 | 50.0 | 52.0 | 12.1 | 25.6 | 61.5 | 35.2 |
| One-Shot RL | 9.8 | 44.7 | 67.2 | 18.3 | 28.6 | 75.5 | 40.7 |
| DoPR | 18.2 | 57.1 | 67.5 | 26.3 | 30.7 | 77.1 | 46.1 |
| *Rollout budget: 30k* | | | | | | | |
| GRPO | 12.3 | 50.0 | 66.1 | 18.0 | 28.3 | 76.1 | 41.8 |
| One-Shot RL | 10.6 | 44.3 | 68.7 | 18.9 | 29.5 | 76.7 | 41.5 |
| DoPR | 18.6 | 57.9 | 74.3 | 30.2 | 33.7 | 80.1 | 49.1 |
| *Rollout budget: 250k* | | | | | | | |
| GRPO | 19.7 | 59.9 | 72.3 | 29.5 | 33.7 | 82.4 | 49.6 |
| One-Shot RL | 13.8 | 54.2 | 66.1 | 19.1 | 30.8 | 77.6 | 43.6 |
| DoPR | 19.7 | 58.3 | 73.5 | 29.2 | 33.1 | 83.1 | 49.4 |

*Table 4.* Ablation study of DoPR. We report Pass@1 after convergence and under a fixed rollout budget.

| Method (Total Rollouts Fixed) | AIME24 | AMC | MATH | MIN. | OLY. | GSM8K | Avg. |
|---|---|---|---|---|---|---|---|
| *Final Performance* | | | | | | | |
| DoPR | 19.7 | 59.3 | 73.5 | 29.4 | 33.9 | 82.2 | 49.6 |
| DoPR-UCB | 19.3 | 59.0 | 73.6 | 29.1 | 33.9 | 81.9 | 49.5 |
| DoPR-None | 18.9 | 58.7 | 73.4 | 29.3 | 33.6 | 80.5 | 49.0 |
| *10k Rollout Budget* | | | | | | | |
| DoPR | 17.4 | 56.2 | 67.0 | 25.9 | 30.2 | 77.3 | 45.7 |
| DoPR-UCB | 16.7 | 55.0 | 66.9 | 25.4 | 29.5 | 76.9 | 45.1 |
| DoPR-None | 10.7 | 40.0 | 66.8 | 24.4 | 30.1 | 77.5 | 41.6 |

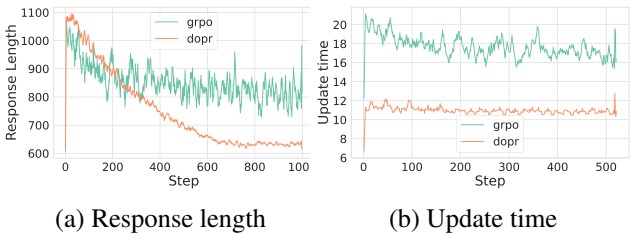

(a) Response length      (b) Update time

*Figure 3.* DoPR consistently yields shorter reasoning trajectories and faster update cycles, indicating improved runtime efficiency and a reduced computational burden for training.

a significantly lower level. This indicates that our method progressively learns to generate more concise reasoning trajectories, which may due to the targeted sample selection and more confident policy updates. In terms of the update time, DoPR consistently achieves lower latency per training step compared to GRPO. This improvement stems from DoPR's single-instance update mechanism, which requires only $G+(K-1)$ rollouts per step rather than the full $K \times G$

rollouts used by GRPO. The update time of DoPR remains stable throughout training, demonstrating its scalability and robustness under fixed compute budgets. Overall, these observations indicate that DoPR not only accelerates convergence but also improves runtime efficiency at both inference and training stages, making it a promising candidate for large-scale deployment in cost-sensitive environments.

## 6. Conclusion

In this work, we present a resource-efficient reinforcement learning framework for training reasoning language models under RLVR. We begin by establishing a theoretical lower bound on the sample complexity of RLVR, providing the first formal understanding of data requirements. Empirically, we show that near-optimal reasoning performance can be achieved with surprisingly few training examples, challenging the prevailing assumption that large-scale datasets are indispensable for RL. In parallel, we propose Dynamic One-Shot Policy Refinement (DoPR), a sampling-efficient algo-

rithm that significantly reduces rollout cost by dynamically selecting a single high-impact sample per batch for policy updates. By combining reward volatility with exploration-aware scoring, DoPR achieves competitive reasoning performance across a suite of mathematical benchmarks, while dramatically reducing computational overhead. Together, our theoretical and algorithmic contributions offer a more sustainable path forward for scaling LLM reasoning training, both in terms of data requirement and compute efficiency.

## Acknowledgement

This work was partially supported by the National Natural Science Foundation of China (Grant No. 62501449). We also thank Zizhe Wang for the valuable discussions and constructive suggestions that helped improve this work.

## Impact Statement

This paper presents work whose goal is to advance the field of machine learning. There are many potential societal consequences of our work, none of which we feel must be specifically highlighted here. We have disclosed all relevant details and ensured research integrity in accordance with the Code of Ethics.

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

# A. Detailed Proof of Theorem 1

For a pre-training policy $\pi_{\theta_0}$, its performance gap with the optimal policy $\pi_{\theta^\star}$ on a reasoning task is:

$$\mathbb{E}_{x\sim\mathcal{D}}[P^{\pi_{\theta^\star}}(x) - P^{\pi_{\theta_0}}(x)] = \epsilon, \tag{21}$$

where $x$ is an instance sampled from the dataset $\mathcal{D}$, and $P^\pi(x)$ denotes the expected verification reward.

**Theorem A.1.** *Consider an optimization procedure where the policy is updated using a single sample at each step, let $\pi_{\theta^\star}$ denotes the optimal policy and $\pi_{\theta_N}$ denotes the policy after $N$ updates. To guarantee that the expected performance gap satisfies*

$$\mathbb{E}_{x\sim\mathcal{D}}[P^{\pi_{\theta^\star}}(x) - P^{\pi_{\theta_N}}(x)] \leq \epsilon', \tag{22}$$

*it suffices that the number of steps $N$ satisfies*

$$N \geq \mathcal{O}(\ln\frac{\epsilon}{\epsilon'}), \tag{23}$$

*where $\epsilon$ denotes the initial performance gap.*

*Proof.* Let $J(\theta) = \mathbb{E}_{x\sim\mathcal{D}}[P^{\pi_\theta}(x)]$ denote the expected performance of the policy parameterized by $\theta$. Assume that $J$ is locally $L$-smooth, i.e., there exists a constant $L > 0$ and a neighborhood $\mathcal{N}$ containing the initial policy $\theta_0$, such that the following Lipschitz inequality holds for all $\theta, \theta' \in \mathcal{N}$:

$$J(\theta') \geq J(\theta) + \langle\nabla J(\theta), \theta' - \theta\rangle - \frac{L}{2}\|\theta' - \theta\|^2. \tag{24}$$

By the policy gradient theorem, $J$'s gradient is given by:

$$\nabla J(\theta) = \mathbb{E}_{x\sim\mathcal{D}}[\nabla_\theta P^{\pi_\theta}(x)]. \tag{25}$$

Assume that the policy is updated at each step via gradient ascent with a fixed learning rate $\alpha > 0$:

$$\theta_{t+1} = \theta_t + \alpha g_t, \tag{26}$$

where $g_t$ is an unbiased stochastic estimate of the true gradient, i.e., $\mathbb{E}[g_t] = \nabla J(\theta_t)$, and the variance of $g_t$ is bounded by $Var(g_t) \leq \delta^2$. Due to the $L$-smoothing,

$$\begin{aligned}
J(\theta_{(t+1)} &\geq J(\theta_t) + \langle\nabla J(\theta_t), \alpha g_t\rangle - \frac{L}{2}\|\alpha g_t\|^2 \\
&= J(\theta_t) + \alpha\langle\nabla J(\theta_t), g_t\rangle - \frac{L\alpha^2}{2}\|g_t\|^2.
\end{aligned} \tag{27}$$

Taking expectations $\mathbb{E}[\cdot|\theta_t]$, we obtain:

$$\begin{aligned}
\mathbb{E}[J(\theta_{t+1}|\theta_t)] &\geq J(\theta_t) + \alpha\|\nabla J(\theta_t)\|^2 - \frac{L\alpha^2}{2}\mathbb{E}[\|g_t\|^2|\theta_t] \\
&\geq J(\theta_t) + \alpha\|\nabla J(\theta_t)\|^2 - \frac{L\alpha^2}{2}(\|\nabla J(\theta_t)\|^2 + \delta^2) \\
&= J(\theta_t) + \alpha(1 - \frac{L\alpha}{2})\|\nabla J(\theta_t)\|^2 - \frac{L\alpha^2\delta^2}{2},
\end{aligned} \tag{28}$$

Pretraining and SFT endow the model with an initial reasoning capability. Coupled with the explicit gradient signal provided by RLVR, the objective $J$ satisfies a local Polyak–Łojasiewicz (PL) condition in the neighborhood induced by pretraining and SFT. Intuitively, the combination of pretraining/SFT (which shapes a favorable local manifold) and the supervised RLVR gradients renders the landscape locally well-conditioned for efficient optimization. That is, $J$ satisfies

$$\|\nabla J(\theta_t)\|^2 \geq c(J(\theta^\star) - J(\theta_t)) = c\Delta_t, \tag{29}$$

for some constant $c > 0$, where $\theta^\star$ is the optimal parameter and $\Delta_t = J(\theta^\star) - J(\theta_t)$ denotes the performance gap at step $t$. Then, we have:

$$\mathbb{E}[J(\theta_{t+1}) - J(\theta_t) \mid \theta_t] \geq \alpha c(1 - \frac{L\alpha}{2})\Delta_t - \frac{L\alpha^2\delta^2}{2}. \tag{30}$$

Let $\eta = \alpha c/2$, and choose $\alpha$ such that $1 - \frac{L\alpha}{2} \geq \frac{1}{2}$, we further require

$$\frac{L\alpha^2\delta^2}{2} \leq \frac{\eta\epsilon'}{2},$$

which leads to the recursive inequality:

$$
\begin{aligned}
&\mathbb{E}\left[\Delta_{t+1} \mid \theta_t\right] \\
&= \Delta_t - \mathbb{E}\left[J(\theta_{t+1}) - J(\theta_t) \mid \theta_t\right] \\
&\leq (1-\eta)\Delta_t + \frac{\epsilon'}{2}.
\end{aligned}
\tag{31}
$$

Let $\eta = \frac{\alpha c}{2}$ and $\frac{L\alpha^2\delta^2}{2} \leq \frac{\eta\epsilon'}{2}$, we can get

$$\mathbb{E}\left[\Delta_{t+1}\right] \leq (1-\eta)\mathbb{E}\left[\Delta_t\right] + \frac{\eta\epsilon'}{2}, \tag{32}$$

If $\Delta_t \geq \epsilon'$, we obtain:

$$\mathbb{E}\left[\Delta_t\right] \leq (1-\eta)^t\Delta_0 + \frac{\epsilon'}{2}, \tag{33}$$

where $\Delta_0 = J(\theta^\star) - J(\theta_0) = \epsilon$ is the initial gap. To ensure $\mathbb{E}\left[\Delta_t\right] \leq \epsilon'$, it suffices that:

$$(1-\eta)^t\epsilon \leq \frac{\epsilon'}{2}. \tag{34}$$

Taking logarithms on both sides yields:

$$t \geq \frac{\ln(\epsilon'/2) - \ln(\epsilon)}{\ln(1-\eta)} = \mathcal{O}(\ln\frac{\epsilon}{\epsilon'}), \tag{35}$$

$\square$

