# OpenReview forum: "Resource-Efficient Reinforcement for Reasoning Large Language Models via Dynamic One-Shot Policy Refinement"
_ICML.cc/2026/Conference — ICML 2026 regular_

### Official Review · Reviewer_VSoc · 2026-03-12

**Soundness:** 3
**Presentation:** 2
**Significance:** 3
**Originality:** 2
**Overall Recommendation:** 4
**Confidence:** 3

**Summary:**

This paper studies how to make RLVR for reasoning LLMs more data- and compute-efficient. The paper has two main claims: first, RLVR may need surprisingly little data to unlock reasoning performance, supported by a simple sample-complexity argument and experiments showing that performance is already close to full-data training with as few as 16 samples; second, the proposed DoPR method improves rollout efficiency by selecting one informative sample per batch using reward variance and an entropy-modulated UCB score, while keeping final accuracy close to GRPO on several math benchmarks. Overall, the paper is about resource-efficient post-training rather than raw reasoning gains.

**Compliance With Llm Reviewing Policy:**

Affirmed.

**Final Justification:**

My concerns have been addressed

**Key Questions For Authors:**

First, how robust is the “16 samples is enough” claim across different data selections and random seeds? Right now it is hard to tell whether this is a stable phenomenon or partly a lucky subset effect. Second, can the authors better justify the local PL assumption, or at least tone down the theoretical claims and present the theorem as a motivating approximation rather than a strong explanation? Third, do the conclusions hold outside math RLVR, for example code, logic, or other verifiable domains? Fourth, since DoPR mainly helps in low-rollout settings, can the authors clarify when one should prefer it over standard GRPO in practice, and whether there are cases where its sample selection becomes too narrow?

**Limitations:**

No, I do not see a clear discussion of the limitations.

**Strengths And Weaknesses:**

**Strength**

The main strength is that the paper asks a useful question that people in RL-for-LLMs care about a lot: not just how to get better performance, but how to do it cheaper. The empirical result that 16 samples already gets close to full-data performance is interesting and, if it holds more broadly, pretty important. The method is also simple and easy to understand: replace full-batch GRPO-style rollout allocation with a one-sample update driven by reward volatility plus an exploration term. Empirically, DoPR looks competitive with GRPO at convergence and clearly stronger under tight rollout budgets, and the ablation suggests the EM-UCB term matters mainly in the low-budget regime, which is the right place to matter.

**Weakness**

My main concern is that the theoretical story feels much stronger than what is actually justified. The bound relies on local smoothness, unbiased gradients, bounded variance, and especially a local PL condition around the RL trajectory; in practice, that is doing a lot of work, so I do not think the theorem really establishes the central empirical claim in a convincing way. Also, the experiments are all on math-style RLVR benchmarks, so it is still unclear whether the “very small data is enough” conclusion generalizes beyond this narrow setting. Another issue is that DoPR’s final performance is mostly at parity with GRPO rather than clearly better, so the contribution is mostly efficiency, not better learning quality. Finally, some of the framing sounds a bit overstated, for example suggesting a fundamental decoupling between reasoning performance and data scale, when the evidence here is still limited to one task family and a few base models.

---

> ### Author Rebuttal · Authors · 2026-03-31
>
> We thank the reviewer for the careful reading and for highlighting both the practical relevance of resource-efficient RLVR and the promise of DoPR. We also appreciate the note that the paper’s main contribution is efficiency-oriented rather than raw performance-oriented. We address the main concerns below.
>
> > Question1: Theoretical story.
>
> **Response1.**
>
> The theorem is intended as a motivating approximation rather than a universal characterization of RLVR. Its purpose is to formalize why, when the base model already possesses a meaningful reasoning prior, a small number of informative updates can suffice to significantly reduce the remaining performance gap. The assumptions including local smoothness, unbiased stochastic gradients, bounded variance, and a local PL-type condition are standard in stochastic optimization. Importantly, these assumptions are required to hold only in the neighborhood of the optimization trajectory around a or instruction-tuned initialization. Moreover, mechanisms such as KL regularization and gradient clipping help keep updates within this local region, making these assumptions more plausible in practice. We therefore view the theorem as a concise explanation of the observed data efficiency, rather than a general claim about all RL landscapes, and will clarify this distinction in the revision.
>
> > Question2: Data scale vs. reasoning performance.
>
> **Response2.**
>
> Our intention is not to claim a complete decoupling between data scale and reasoning performance. The main point is narrower: RLVR can often reach near-saturated reasoning performance with a much smaller number of informative samples than one might expect. This is also consistent with existing few-shot RLVR results. For instance, One-Shot RL shows that meaningful reasoning improvements can be obtained from a single sample. Our work extends this observation by showing that similar phenomenon can be captured more systematically through both theory and experiments. At the same time, the conclusion depends on the base model and the quality of the initialization, we will state more carefully in the revised paper.
>
> > Question3: Final performance of DoPR.
>
> **Response3.**
>
> DoPR is a rollout-efficient variant of GRPO, so it is not meant to fundamentally change the asymptotic performance ceiling of RLVR. In reasoning model training, most domain knowledge is acquired during pre-training and SFT, while RL mainly reshapes the output distribution and aligns the model toward higher-reward trajectories. For this reason, DoPR is expected to achieve performance that is comparable to GRPO at convergence. The main benefit of DoPR lies in how quickly that performance can be reached. By dynamically selecting the most informative sample in each mini-batch, DoPR substantially reduces rollout cost. This is why its advantage is most evident under constrained budgets, as shown in Table 3: when rollout budget is limited, DoPR reaches strong performance much earlier than GRPO. The goal of DoPR is to make RLVR more resource-efficient, not to redefine the final performance upper bound.
>
> > Question4: The robustness of 16 samples.
>
> **Response4.**
>
> We thank the reviewer for the constructive feedback. Our reported results are based on the average of 5 independent runs, each with a different randomly sampled training subset. We also provide the corresponding standard deviations alongside the average performance (see details in the response to Question 1 of Reviewer 2). The results exhibit low variance across runs, indicating that this is a stable phenomenon rather than a lucky subset effect.
>
> > Question5: Results on other tasks.
>
> **Response5.**
>
> We extend the evaluation to non-math tasks by training DoPR on math data and then evaluating the resulting model on HumanEval for coding and BBH for logic reasoning, using the Qwen2.5-1.5B backbone, and the results are shown below. We observe consistent gains on both out-of-domain benchmarks. This suggests that DoPR does not merely overfit to math training data, but can still improve the underlying reasoning policy in a broader sense. The results provide additional evidence that the proposed method has some degree of transferability beyond the training distribution. We will include these results in the revised version.
> |Method|HumanEval|BBH|
> |-----|-----|-----|
> |Base|37.2|45.1|
> |DoPR|39.1|46.7|
>
> > Question6: When to prefer DoPR.
>
> **Response6.**
>
> DoPR is most useful when rollout budget or training time is limited, since it reduces the number of rollouts and therefore reaches strong performance with less compute. Importantly, DoPR does not depend on a fixed small training subset. It can be applied to either a reduced dataset or the full dataset, because the key operation happens at the batch level by selecting the most informative sample for optimization. This means the selection process does not become narrowly tied to a specific subset of samples over training; rather, it adapts dynamically as the model improves.

---

> > ### Author Rebuttal · Reviewer_VSoc · 2026-04-05
> >
> > Thanks for the reply. My concerns have beed addressed. I will raise my score.

---

### Official Review · Reviewer_ukHn · 2026-03-12

**Soundness:** 3
**Presentation:** 3
**Significance:** 3
**Originality:** 3
**Overall Recommendation:** 4
**Confidence:** 4

**Summary:**

This paper proposes DoPR, an algorithm which selects a single high-value instance per mini-batch based on a score. The score is computed based on (1) the exponential moving average of the variance of past rewards for that sample and (2) an EM-UCB exploration term. DoPR achieves comparable performance to GRPO while using significantly fewer samples.

**Compliance With Llm Reviewing Policy:**

Affirmed.

**Final Justification:**

I would like to maintain my positive rating, but I did not further raise my score as fundamentally I feel that such methods which introduce complexity in order to reduce computation can be limited in their impact.

**Key Questions For Authors:**

See weaknesses

**Limitations:**

Yes

**Strengths And Weaknesses:**

Strengths:
- Reducing computational costs of RLVR is timely and relevant
- Substantial efficiency gains (10k rollouts vs 50k for GRPO)
- The method is simple to integrate into existing RLVR pipelines with minimal added complexity
- Experiments span multiple model families, data scales, and rollout budgets

Weaknesses:
- The results in Table 2 could be an artifact of Qwen2.5 training, so ideally this would be shown also on other model families
- The choice of scoring function seems a bit arbitrary and not well-ablated. There are many potential ways to measure how informative a sample is (e.g., various ways to measure uncertainty or measuring gradient norms) so the summation of these two terms in particular is not well-motivated

---

> ### Author Rebuttal · Authors · 2026-03-31
>
> We thank the reviewer for the positive assessment of the efficiency gains, the simplicity of DoPR, and the breadth of the experiments. We respond to the two concerns below.
>
> > Question1: Results on other model families.
>
> **Response1.**
>
> We thank the reviewer for this suggestion. To verify that the data scaling findings in Table 2 are not an artifact of Qwen2.5, we conduct the same experiment on LLaMA-3.1-8B, which is a different backbone family and a more general model. As shown below, the trend is consistent: 16 samples suffice for near-optimal performance, confirming that this is a general property of RLVR rather than model-specific. This additional result supports the main claim that the sample-efficiency trend is not tied to a particular backbone family. Instead, it is consistent with our theoretical analysis: once the base policy already has a reasonable reasoning prior, RLVR can be effective with a surprisingly small number of informative samples.
>
> | Method | AIME24 | AMC | MATH | MIN. | OLY. | GSM8K | Avg. |
> |--------|--------|-----|------|------|------|-------|------|
> | GRPO (Full) | 3.3 | 10.0 | 25.8 | 11.4 | 6.5 | 70.9 | 21.3 |
> | GRPO (16)   | 3.3 | 9.2  | 24.6 | 10.8 | 6.1 | 69.7 | 20.6 |
> | DoPR (16)   | 3.3 | 15.0 | 28.0 | 16.9 | 7.7 | 72.8 | 24.0 |
> | DoPR (8)    | 0.0 | 11.3 | 25.1 | 13.2 | 5.9 | 70.2 | 20.9 |
>
> > Question2: The choice of scoring function.
>
> **Response2.**
>
> The scoring function in DoPR is designed to be a minimal acquisition rule that combines two complementary signals already available from lightweight rollouts: 1) reward variance, which reflects whether a sample is currently unstable and therefore potentially informative for policy refinement; 2) entropy-modulated UCB, which encourages revisiting under-updated samples only when the policy remains uncertain.
>
> The reward-variance term captures the exploitation side of sample utility: samples whose outcomes fluctuate across rollouts are exactly the ones where the policy still has room to improve. The entropy-gated UCB term captures the exploration side: among samples that have not been updated frequently, exploration is only amplified when the model still exhibits high uncertainty, which avoids wasting rollouts on samples that are already confidently solved. In this sense, the two terms play clearly different roles, and their combination is meant to reflect the two main factors that determine whether a sample is worth spending additional rollout budget on.
>
> We also chose this formulation because it is lightweight and training-aware. More elaborate alternatives, such as gradient-norm-based scoring or learned acquisition models, would either require extra backward passes, additional rollouts, or another trainable module, which would undermine the main goal of DoPR: reducing rollout cost while keeping the method easy to integrate into existing RLVR pipelines. By contrast, our score uses only statistics that are already produced during training, and therefore adds negligible overhead.
>
> Finally, our existing ablation already supports this design: removing the UCB component or replacing the selection rule with random sampling reduces efficiency under fixed rollout budgets, which indicates that the score is not an arbitrary heuristic but a practical decomposition of sample utility into complementary uncertainty and exploration signals. We will further clarify this motivation in the revision.

---

> > ### Author Rebuttal · Reviewer_ukHn · 2026-04-01
> >
> > Thanks to the authors for addressing my concerns. I would like to maintain my positive rating.
> >
> > (I did not further raise my score as fundamentally I feel that such methods which introduce complexity in order to reduce computation can be limited in their impact).

---

### Official Review · Reviewer_aJWW · 2026-03-14

**Soundness:** 3
**Presentation:** 3
**Significance:** 3
**Originality:** 3
**Overall Recommendation:** 4
**Confidence:** 3

**Summary:**

This paper studies how to reduce the data and cost of RLVR when training reasoning LLMs. The authors derive a theoretical bound and show that RLVR training can converge with relatively few updates. Motivated by this observation, the paper proposes Dynamic One-Shot Policy Refinement (DoPR), a training strategy that selects a single informative sample from each batch for policy updates instead of performing full rollouts on every sample.

Experiments show that DoPR can achieve performance comparable to GRPO while significantly reducing rollout cost.

**Compliance With Llm Reviewing Policy:**

Affirmed.

**Key Questions For Authors:**

- how do the author might explain on the diminishing return on the performance gain in table 3?
- it is interesting to see the curve on response length. does authors have any more explanations on this?

**Limitations:**

yes

**Strengths And Weaknesses:**

strength:
- empirical success
- match GRPO performance will less rollout compute


weakness:
- the main thm is fairly general and trivial: this is essentially a textbook proof of convergence for stochastic gradient descent. The assumption is quite similar: The objective function is locally L-smooth; the variance of the stochastic gradient is bounded; the objective satisfies a local PL condition. It is hard to convince the readers this is very specific to the reasoning scenario and RLVR. The backing of these assumptions is quite shallow. For example: "We remark that this assumption holds primarily in the vicinity of the optimization trajectory, and its validity is empirically ensured by conservative step sizes, KL regularization, and gradient clipping.", which lacks theoretical and empirical evidence in practice.
- missing baselines: while I like the idea of the proposed method, I think the authors might miss some naive baselines. For example, those difficulty-based or entropy-based methods, which can also be applied to select that single data in each round to get full rollouts.
- I would suggest better organization on the main table 1: while it is surprising to match GRPO with less compute, the current visualization is hard to observe that. It should be easier for readers to comprehend if the rows are grouped by base models.
- some quantitive analysis might be helpful: since there is only one data used in each batch, it will be interesting to see the property of that single sample over the training process. For example, by tracking its difficulty, type of question etc.

---

> ### Author Rebuttal · Authors · 2026-03-31
>
> We thank the reviewer for the thoughtful comments and recognizing both the empirical strength of DoPR and the practical relevance of reducing rollout cost in RLVR. We respond to the concerns below.
>
> > Question1: Theoretical and empirical evidence.
>
> **Response1.**
>
> We thank the reviewer for the rigorous evaluation of our theoretical framework. We clarify that Theorem 3.1 is not intended to introduce a fundamentally novel proof, but to provide a principled abstraction explaining a counterintuitive empirical phenomenon: why extreme data efficiency is possible in RLVR. The theorem characterizes a local optimization regime where the model starts from an SFT checkpoint with non-trivial reasoning ability, the reward is bounded and explicit, and updates are conservative due to standard mechanisms such as KL regularization and gradient clipping. Under these conditions, a local smoothness and gradient-dominance view serves as a reasonable approximation of the optimization dynamics. Importantly, these assumptions are inherent to the GRPO framework on which DoPR is built. These insights are also supported by empirical evidence: training with as few as 16 samples achieves performance close to the full-data setting.
>
> > Question2: Additional baselines.
>
> **Response2.**
>
> We thank the reviewer for this comment. We compare DoPR with SEED-GRPO[1], a entropy-based method, and AdaRFT[2], a difficulty-based method. We report results on Qwen2.5-Math-1.5B. Note that both baselines use significantly more training data, while DoPR achieves competitive or superior performance with only 16 samples, further demonstrating its efficiency.
>
> |Method|Training Data|AIME24|AMC|MATH|MIN.|OLY.|Avg.|
> |-------|------|-----|-----|-----|-----|-----|-----|
> |Base|—|16.7|43.4|61.8|15.1|28.4|33.1|
> |AdaRFT|10000|12.1|57.5|66.2|14.3|21.9|34.4|
> |SEED-GRPO|8500|23.3|50.6|75.4|26.8|41.3|43.5|
> |GRPO|1209|20.0|60.0|71.8|29.8|34.2|43.2|
> |DoPR|16|19.7|59.3|73.5|29.4|33.9|43.2|
>
> [1] Seed-grpo: Semantic entropy enhanced grpo for uncertainty-aware policy optimization.
>
> [2] Efficient reinforcement finetuning via adaptive curriculum learning.
>
> > Question3: Organization of table 1.
>
> **Response3.**
>
> Thanks for this suggestion. We will reorganize Table 1 by grouping results according to the base model, and highlight the best and second-best entries more explicitly.
>
> > Question4: Quantitive analysis per batch.
>
> **Response4.**
>
> We thank the reviewer for this insightful suggestion. From a within-batch perspective, we observe that samples of moderate difficulty are more likely to be selected. These examples are typically those on which the model exhibits mixed outcomes across rollouts, because such samples tend to have higher reward variance, making them more informative. Since the training samples are randomly sampled from the dataset, we do not impose any fixed preference over question types. From a training-dynamics perspective, the selected samples tend to become progressively harder over time. This is because, as the model improves, easier instances are answered correctly more consistently. In contrast, the remaining samples with higher uncertainty gradually dominate the acquisition score. Moreover, among samples with similar difficulty, those that have been selected fewer times are more likely to be chosen.
>
> > Question5: Diminishing return on the performance gain.
>
> **Response5.**
>
> The diminishing return observed is expected and consistent with the role of DoPR. When the rollout budget is small, DoPR has a clear advantage because it concentrates computation on the most informative sample in each batch, so each rollout contributes more directly to policy improvement. As the budget increases, GRPO also has more opportunities to refine the policy, and both methods gradually approach their convergence regime. The marginal gain from additional rollouts becomes smaller, which naturally reduces the gap between methods. In other words, the main benefit of DoPR is not to change the final optimization target, but to reach a strong solution with fewer rollouts.
>
> > Question6: Response length.
>
> **Response6.**
>
> We attribute the declining response length of DoPR to its targeted sample selection mechanism. Since DoPR consistently directs the full rollout budget toward the most informative sample at each step, the policy receives focused, high-signal updates. This allows the model to converge faster toward confident reasoning patterns, progressively eliminating redundant exploration steps. That is, the model learns to solve problems more directly. Importantly, this length reduction does not come at the cost of accuracy, DoPR maintains competitive performance across all benchmarks. This suggests that the model is learning a more efficient reasoning distribution, producing concise yet correct solutions rather than verbose trial-and-error traces. We view this as an additional practical benefit of DoPR: shorter responses reduce inference cost in deployment.

---

> > ### Author Rebuttal · Reviewer_aJWW · 2026-04-04
> >
> > thanks for the rebuttal. I will maintain my positive score.

---

### Official Review · Reviewer_CBdi · 2026-03-18

**Soundness:** 2
**Presentation:** 2
**Significance:** 1
**Originality:** 2
**Overall Recommendation:** 2
**Confidence:** 4

**Summary:**

The paper introduces Dynamic One-Shot Policy Refinement (DoPR), a method designed to mitigate the heavy data and computational costs associated with Reinforcement Learning with Verifiable Rewards (RLVR) in reasoning LLMs. The authors argue that reasoning capabilities can be unlocked with an extremely minimal data regime—as few as 16 samples. To reduce computational overhead, DoPR uses an Entropy-Modulated Upper Confidence Bound (EM-UCB) scoring mechanism to dynamically select a single, highly informative sample per mini-batch for policy updates. This strategy reduces the number of required rollouts by nearly an order of magnitude compared to standard GRPO , while maintaining competitive Pass@1 accuracy on standard mathematical reasoning benchmarks.

**Compliance With Llm Reviewing Policy:**

Affirmed.

**Key Questions For Authors:**

See weakness.

**Limitations:**

No.

**Strengths And Weaknesses:**

Strengths:
- The paper correctly targets the massive computational overhead of group-wise rollouts in algorithms like GRPO, which is a significant pain point for scaling RLVR.
- The DoPR algorithm itself—using running momentum of reward variance and policy entropy (EM-UCB) to selectively allocate rollout budgets—is a clever and lightweight heuristic for filtering out redundant training updates.
- Even if slightly misrepresented, the paper provides strong empirical evidence that RLVR in modern post-training acts primarily as a "capability activator" rather than a "performance booster," heavily relying on the pre-trained knowledge base.

Weakness:
- The claims and conclusion could be misleading:
The paper claims strong performance using only 16 samples but fails to report any variance, standard deviation, or multiple-seed averages. In extreme low-resource RL, this masks the inherent instability of the method, as the gradient direction is highly susceptible to the specific heuristics of those 16 questions. Also, the success of the 16-sample regime relies entirely on the fact that the chosen base models (like Qwen2.5-Math) are already heavily supervised and fine-tuned for mathematics. The RL process is merely aligning the output format to the reward signal, not teaching the model how to reason from scratch

- Overfitting concern
The training setup runs 1000 optimization steps over just 16 samples. This is fundamentally forcing the model to overfit to the specific reasoning trajectory formats (e.g., verifiable reward structures) of a tiny dataset, rather than learning generalized, out-of-distribution mathematical reasoning.

- Even though the paper's motivation is about the computational cost. Limiting the training to 16 samples fails to improve the cost. Your RL experiments still runs for 1000 steps.

---

> ### Author Rebuttal · Authors · 2026-03-31
>
> We sincerely thank the reviewer for the thoughtful comments. We appreciate the recognition of the paper’s motivation, the efficiency perspective of DoPR, and the empirical evidence suggesting that RLVR can act as a capability activator rather than merely a performance booster. We address the concerns point by point below.
>
> > Question1: Variance and standard deviation.
>
> **Response1:**
>
> We thank the reviewer for the constructive feedback. We would like to clarify that our reported results were already based on the average of 5 independent runs, each using a different randomly selected training subset. We include standard deviation alongside the average performance below. The results indicate that while the standard deviation exhibits a slight increase as the sample size decreases, the overall training process remains consistently stable.
> |Method|AIME24|AMC|MATH|MIN.|OLY.|GSM8K|Avg|
> |--------|-----|-----|-----|-----|-----|-----|-----|
> |GRPO 128 samples|20.0|57.5|71.7|29.8|32.1|82.7|48.9±1.47|
> |GRPO 16 samples|19.3|58.5|69.8|28.2|32.9|81.5|48.4±2.68|
> |GRPO 8 samples|10.0|50.0|69.8|25.9|31.3|77.0|44.0±3.30|
> |DoPR 128 samples|19.9|60.1|73.0|27.8|32.7|82.1|49.3±1.75|
> |DoPR 16 samples|19.7|59.3|73.5|29.4|33.9|82.2|49.6±2.86|
> |DoPR 8 samples|9.3|44.7|68.9|26.4|30.8|77.4|42.9±3.14|
>
> > Question2: Choice of base model.
>
> **Response2:**
>
> We appreciate the reviewer's perspective, which touches upon the fundamental distinction between knowledge acquisition and capability activation in large language models.
>
> * It is a well-established paradigm that domain knowledge is primarily acquired during the pre-training and SFT stages, and RLVR acts as a process of refining and activating pre-existing capabilities. The objective of our research is to investigate the data efficiency of this phase, which further aligns the policy with explicit reward signals, improve reasoning behavior. Our finding that only 16 samples can effectively "activate" these capabilities provides a new perspective on the minimal supervision required for RL.
>
> * Within this context, math-specialized base models are commonly used in RLVR because they already possess non-trivial mathematical competence, which makes it easier to isolate the effect of the RL algorithm itself and reduces confounding from domain learning during SFT.
>
> * Our experiments are not limited to math-specialized models. As shown in Table 1, we report results on LLaMA-8B, a general-purpose backbone. In addition, we further include results on Qwen2.5-base 1.5B below. These results show that DoPR remains effective on more general base models as well.
> |Method|AIME24|AMC|MATH|MIN.|OLY.|GSM8K|Avg|
> |---------|-----|-----|-----|-----|-----|-----|-----|
> |Base|0.0|0.0|3.0|2.2|2.4|3.9|1.9|
> |GRPO|4.3|31.1|53.2|17.4|18.6|64.8|31.6|
> |One-Shot RL|1.2|13.8|42.9|13.5|17.4|59.5|24.7|
> |DoPR|4.6|30.3|55.4|17.1|18.5|62.7|31.4|
>
> > Question3: Overfitting concern.
>
> **Response3:**
>
> * Different from standard supervised learning, RLVR utilizes a verifier-defined reward signal. During the 1000 optimization steps, the model explores reasoning trajectories that satisfy the logic of the verifier. The "16 samples" represent 16 reasoning environments, and the RL process optimizes the policy’s ability to navigate these environments toward a verifiable solution.
>
> * As established in Theorem 3.1, the sample requirement for convergence is dictated by the initial performance gap. Given that our base models already possess latent reasoning capabilities, the RL stage acts as a capability activator. A small number of samples is sufficient to steer the model toward a more efficient reasoning mode that already exists within its parameters.
>
> * The results further support this interpretation. Even the One-Shot RL baseline shows with a carefully chosen sample, the model can make meaningful updates. We evaluate all models on multiple benchmarks covering diverse problems, including difficulty levels, problem types (number, expression, equation), and domains (mathematics and physics). The consistent improvements across these settings indicate that DoPR enhances the underlying reasoning policy.
>
> > Question4: Computational cost.
>
> **Response4.**
>
> DoPR improves efficiency by reducing the rollout complexity for training. For standard GRPO, if a batch contains K samples and each sample requires G rollouts, then each step incurs K×G rollouts. By contrast, DoPR first performs a single rollout for each sample, uses the historical reward statistics to select the most informative instance, and then completes the remaining rollouts for it. As a result, the rollout cost is reduced to G+K−1. This reduction directly translates into lower training-time compute. As shown in Table 3, this efficiency gain is reflected empirically: DoPR with a 30k rollout budget achieves performance comparable to GRPO with a 250k rollout budget. This supports our main claim that DoPR improves rollout efficiency without sacrificing final reasoning performance.

---

> > ### Author Rebuttal · Reviewer_CBdi · 2026-04-03
> >
> > Thank you for your detailed response. What happens if you have, say, 10k samples? Training with 1k optimization steps is already a computational-heavy setup. In this scenario, I don't think sample size matters.

---

> > > ### Author Response · Authors · 2026-04-04
> > >
> > > We thank the reviewer for the constructive follow-up. We are glad that part of the concerns have been addressed. Below, we provide a more detailed clarification regarding the remaining question on computational cost.
> > >
> > > * We would like to clarify that our paper addresses data efficiency and compute efficiency as two separate contributions. The 16-sample finding demonstrates that RLVR requires minimal data to activate reasoning capabilities, which reduces the cost of data collection and curation rather than training compute. It does not claim that the number of optimization steps can be reduced proportionally. In contrast, the compute saving comes from DoPR's rollout reduction: each training step costs G+(K-1) rollouts instead of K x G, regardless of the dataset size. Therefore, even when the number of optimization steps is fixed, DoPR still substantially lowers wall-clock training cost. In fact, with a larger dataset like 10k samples, the per-step savings from DoPR become even more significant, as a larger batch size K leads to a greater gap between K x G and G+(K-1). These two contributions are complementary: fewer samples to collect, and less compute per step to train.
> > >
> > > * As reported in our response to Q4 to Reviewer #1, DoPR reduces total GPU hours from 187.4 to 92.3 (>50% reduction) with comparable accuracy on the Qwen2.5-7B model. This confirms that the per-step rollout savings translate to substantial wall-clock improvement in practice, even though the speedup is not perfectly proportional to the rollout reduction due to communication overhead and other system-level costs.
> > >
> > > * Regarding the maximum training step budget, 1000 steps is merely a termination cap, not the number of steps required for convergence. This setting follows the protocol in [1], which uses maximum step budgets of 1000 and 2000 for 1.5B and 7B models, respectively. In our experiments, we found that both DoPR and GRPO converge substantially earlier than this upper bound. We therefore set the maximum step budget to 1000 to allow the converged model to remain in a stable regime for a period of time, which helps rule out spurious convergence. We report MATH500 Pass@1 of DoPR on Qwen2.5-Math-1.5B and 7B at different training steps below, and the results show that DoPR already converges at around 400-500 steps, indicating that the actual compute can be further reduced by early stopping.
> > >
> > > | Step | 100 | 200 | 300 | 400 | 500 | 600 | 800 | 1000 |
> > > |------|-----|-----|-----|-----|-----|-----|-----|------|
> > > | DoPR (1.5B) | 58.2 | 68.4 | 71.0 | 72.7 | 72.9 | 73.0 | 73.2 | 73.5 |
> > > | DoPR (7B) | 62.4 | 70.2 | 72.6 | 73.2 | 73.4 | 73.6 | 73.8 | 73.8 |
> > >
> > > Overall, although DoPR does not reduce the number of optimization steps, it significantly lowers the rollout cost per step, thereby reducing total training time by over 50% while maintaining performance comparable to GRPO.
> > >
> > > [1] Wang Y, Yang Q, Zeng Z, et al. Reinforcement learning for reasoning in large language models with one training example[J]. arXiv preprint arXiv:2504.20571, 2025.

---

### Official Review · Reviewer_MX8H · 2026-03-20

**Soundness:** 3
**Presentation:** 3
**Significance:** 3
**Originality:** 3
**Overall Recommendation:** 4
**Confidence:** 3

**Summary:**

This paper aims to solve the problems about high rollout cost and large-scale dataset requirements of RLVR (Reinforcement Learning with Verifiable Rewards). To tackle these problems, this paper first proposes a theoretical lower bound on the sample complexity to demonstrate the effectiveness of small scale training datasets to achieve near-optimal reasoning performance. Based on the findings, the authors propose DoPR (Dynamic One-Shot Policy Refinement), a RL training strategy that discards the high cost exploration for models during training while maintain the performance. The paper also indicates a interesting observation that RL supervision serves more as a capability activator rather than a performance booster.

**Compliance With Llm Reviewing Policy:**

Affirmed.

**Final Justification:**

Author's rebuttal has addressed my concerns. However, considering the DoPR still have some limitations like generalizability (improvement on other data sets is marginal compared to the base model), and I’m still doubt whether the such small number of samples can impact larger scales of models. So I tend to maintain my score rather than raise it.

**Key Questions For Authors:**

1. The limitations about the DoPR are not fully discussed in the paper. I wonder if such a small number of samples can be effective to larger scale of models, like 32B or 70B? and can DoPR also have good performance in other areas like coding?
2. The paper mentions that the base model is Qwen-Math, which has been previously trained by sufficient math problems. Whether DoPR can perform well on other base models that not very domain specific?
3. Is there any statistics about actual training time like GPU hours? Can author provide some quantitative analysis on the efficiency between DoPR and GRPO?

**Limitations:**

The limitations of the DoPR are not fully discussed in the paper, but the societal impact have clearly written in the paper.

**Strengths And Weaknesses:**

Strengths:
1. The paper gives a interesting and special insight about the current RL training method, and presents a comprehensive and detailed analysis on the proposed idea.
2. The proposed method DoPR is well-structured and easy to understand.
3. The experiments are solid and thorough across multiple models and benchmarks. DoPR gains nearly equal and even better performance compared to full-GRPO training while using much smaller samples, which shows the effectiveness of DoPR.

Weaknesses:
1. Some statistics in the table are not very clear. It would be better if the table highlights the best and second performance, and the comparison between different models of the same parameter in Table.1 can be put together to show better comparison.
2. The scalability and generalizability of DoPR should be further discussed. The experiments in the paper only conducted on math datasets, and the parameters of the models are between 1.5B and 8B. But there are also other areas like coding and logic reasoning problems, and larger model scales like 32B and 70B. These are not discussed in the paper.

---

> ### Author Rebuttal · Authors · 2026-03-31
>
> We sincerely thank the reviewer for the careful reading and positive assessment of our work. We especially appreciate the recognition of our theoretical insight, the clarity of DoPR, and the thorough experimental analysis. We address the concerns point by point below.
>
> > Question1: Table presentation.
>
> **Response1:**
>
> We thank the reviewer for the constructive suggestion. In the revised version, we will restructure the experimental tables to enhance readability by: (1) explicitly highlighting the best and second-best results; and (2) grouping models by parameter scale to facilitate a more direct comparison between different backbones.
>
>
> > Question2 & 3: Scalability and generalizability of DoPR.
>
> **Response2 & 3:**
>
> We thank the reviewer for raising this important point. Regarding larger model scales (e.g., 32B or 70B), we acknowledge that due to current computational constraints, we are unable to conduct full RL training experiments. We will explicitly clarify this limitation in the revised version. We emphasize that our findings are fundamentally scale-agnostic. Theoretically, the convergence behavior depends on gradient dynamics and the initial performance gap rather than parameters. Empirically, we observe consistent trends across multiple scales (1.5B to 7B), where reasoning performance remains insensitive to dataset size. These results suggest that the data and rollout efficiency of DoPR is a structural property of the optimization process itself, rather than a phenomenon limited to smaller models. Once sufficient computational resources become available, we plan to systematically investigate scaling behavior.
>
> As for other tasks, we extend the evaluation to non-math tasks to better examine the generalization of our approach beyond mathematical reasoning, and the results are shown below. In this experiment, we evaluate our DoPR-trained model (trained on math data) on HumanEval (coding) and BBH (logic reasoning) using the Qwen2.5-1.5B backbone. It can be observed that DoPR brings modest gains on these out-of-domain benchmarks, indicating that it does not overfit to math tasks. We will include these results in the revised version.
>
> |Model|Method|HumanEval|BBH|
> |-------|--------|-----------|-----|
> |Qwen2.5-1.5B|Base|37.2|45.1|
> |Qwen2.5-1.5B|DoPR|39.1|46.7|
>
> > Question4: Dependence on the base model.
>
> **Response4:**
>
> We thank the reviewer for this insightful comment. We clarify that while Qwen-Math serves as a strong baseline, the efficacy of DoPR is not limited to domain-specific models.
>
> * Using math-oriented models is a standard paradigm in current RLVR research [1,2,3]. Mathematical tasks provide unambiguous, verifiable ground truths, which allow us to isolate the performance gains of the RL refinement process from the interference of SFT quality. This ensures that the observed improvements are strictly attributed to the DoPR algorithm rather than the internal knowledge of the backbone.
> * To demonstrate the robustness of DoPR across different architectures, we have included results for Llama-3-8B in Table 1. As a general-purpose backbone, it demonstrates consistent performance gains under DoPR, confirming that our method's advantages are not confined to specialized "Math" series models.
> * To further address this concern, we conducted additional experiments on Qwen2.5-base 1.5B, a versatile pre-trained model without math-specific tuning. The results shown below reveal that DoPR achieves reasoning accuracy comparable to GRPO while requiring significantly lower computational overhead. We will include these analyses in the revised manuscript.
>
> |Method|AIME24|AMC|MATH|MIN.|OLY.|GSM8K|Avg|
> |---------|--------|-----------|-----|-----|-----|-----|-----|
> |Qwen2.5-base 1.5B|0.0|0.0|3.0|2.2|2.4|3.9|1.9|
> |Qwen2.5-base 1.5B GRPO|4.3|31.1|53.2|17.4|18.6|64.8|31.6|
> |Qwen2.5-base 1.5B One-Shot RL|1.2|13.8|42.9|13.5|17.4|59.5|24.7|
> |Qwen2.5-base 1.5B DoPR|4.6|30.3|55.4|17.1|18.5|62.7|31.4|
>
> [1] Seed-grpo: Semantic entropy enhanced grpo for uncertainty-aware policy optimization.
> [2] Deepseek-r1: Incentivizing reasoning capability in llms via reinforcement learning.
> [3] Understanding r1-zero-like training.
>
> > Question5: Training efficiency and GPU hours.
>
> **Response5:**
>
> We thank the reviewer for this suggestion. We report the actual training cost on A800 GPUs for the Qwen2.5-7B model (1000 steps). With K=16 and G=16, DoPR reduces per-step rollouts from KxG=256 to G+(K-1)=31 (87.9% reduction), which will speed up training time. Furthermore, as shown in Table 3, DoPR with only 10k total rollouts achieves comparable accuracy to GRPO with 250k rollouts, demonstrating ~25x better rollout utilization.
>
> |Method|Rollouts/Step|GPU Hours|Avg. Acc|
> |--------|--------------|------------------|----------|
> |GRPO|256|187.4|54.1|
> |DoPR|31|92.3|53.1|

---

> > ### Author Rebuttal · Reviewer_MX8H · 2026-04-02
> >
> > Thank authors for the detailed rebuttal. My score will stay positive.

---

### Decision · Program_Chairs · 2026-04-30

**Decision:**

Accept (regular)

**Comment:**

The paper introduces DoPR, a method that reduces the sample size of RL training. The authors identified that the models can use much fewer examples to achieve the same performance for RL. The authors established theoretical insights and provided comparisons with prior methods under different rollout budgets. The experiments are conducted on Math models and general models. As a result, 50% of the compute could be saved. The reviewers lean towards acceptance with one reviewer having concerns on the optimization steps. The authors provided justifications on the specific settings and the overall compute being saved. I feel that the clarification makes sense.